# Prevention of DNA Replication Stress by CHK1 Leads to Chemoresistance Despite a DNA Repair Defect in Homologous Recombination in Breast Cancer

**DOI:** 10.3390/cells9010238

**Published:** 2020-01-17

**Authors:** Felix Meyer, Saskia Becker, Sandra Classen, Ann Christin Parplys, Wael Yassin Mansour, Britta Riepen, Sara Timm, Claudia Ruebe, Maria Jasin, Harriet Wikman, Cordula Petersen, Kai Rothkamm, Kerstin Borgmann

**Affiliations:** 1Laboratory of Radiobiology and Experimental Radiooncology, Center of Oncology, University Medical Center Hamburg-Eppendorf, 20246 Hamburg, Germany; fe.meyer@uke.de (F.M.); saskiaalexandra.becker@uke.de (S.B.); sandra.classen@uke.de (S.C.); a.parplys@uke.de (A.C.P.); w.mansour@uke.de (W.Y.M.); riepen@uke.de (B.R.); k.rothkamm@uke.de (K.R.); 2Tumor Biology Department, National Cancer Institute, Cairo University, Cairo 11796, Egypt; 3Department of Radiation Oncology, Saarland University, 66421 Hamburg/Saar, Germany; sara.timm@uks.eu (S.T.); claudia.ruebe@uks.eu (C.R.); 4Developmental Biology Program, Memorial Sloan Kettering Cancer Center, New York, NY 10065, USA; m-jasin@ski.mskcc.org; 5Department of Tumor Biology, University Center Hamburg-Eppendorf, 20246 Hamburg, Germany; h.wikman@uke.de; 6Department of Radiotherapy and Radiooncology, University Medical Center Hamburg-Eppendorf, 20246 Hamburg, Germany; cor.petersen@uke.de

**Keywords:** triple-negative breast cancer (TNBC), chromosomal instability (CIN), CIN70 score, homologous recombination (HR), DNA-damage response (DDR), CHK1

## Abstract

Chromosomal instability not only has a negative effect on survival in triple-negative breast cancer, but also on the well treatable subgroup of luminal A tumors. This suggests a general mechanism independent of subtypes. Increased chromosomal instability (CIN) in triple-negative breast cancer (TNBC) is attributed to a defect in the DNA repair pathway homologous recombination. Homologous recombination (HR) prevents genomic instability by repair and protection of replication. It is unclear whether genetic alterations actually lead to a repair defect or whether superior signaling pathways are of greater importance. Previous studies focused exclusively on the repair function of HR. Here, we show that the regulation of HR by the intra-S-phase damage response at the replication is of overriding importance. A damage response activated by Ataxia telangiectasia and Rad3 related-checkpoint kinase 1 (ATR-CHK1) can prevent replication stress and leads to resistance formation. CHK1 thus has a preferred role over HR in preventing replication stress in TNBC. The signaling cascade ATR-CHK1 can compensate for a double-strand break repair error and lead to resistance of HR-deficient tumors. Established methods for the identification of HR-deficient tumors for Poly(ADP-Ribose)-Polymerase 1 (PARP1) inhibitor therapies should be extended to include analysis of candidates for intra-S phase damage response.

## 1. Introduction

Breast cancer is a very heterogeneous disease whose prognosis is determined by the molecular subtype. Tumors of the triple-negative breast cancer (TNBC) subtype show the worst prognosis [1], which could be attributed to increased chromosomal instability (CIN) [2]. Using the CIN, a gene expression profile consisting of 70 genes associated with a functional aneuploidy (CIN70 score) was extracted. The CIN70 score is the highest for TNBCs compared to other subtypes and a correlation of high/intermediate CIN70 score with prognosis has been observed in numerous studies and tumor entities [3]. A high CIN70 value may be caused by defects in DNA repair, as all BRCA1/2 deficient tumors and one in four sporadically occurring TNBCs show a defect in homologous recombination-mediated (HR-mediated) DNA double-strand break (DSB) repair. HR represents the main DSB repair pathway in the S-phase [4]. It processes DSBs with two open ends, one-ended replication-associated DSB and stalled DNA replication forks.

Two-ended DSBs are repaired by synthesis-dependent strand annealing (SDSA) [5]. As first step DNA end resection is initiated via the nucleases CtIP and MRE11 and is completed by DNA2, EXO1 and BLM helicase [6]. RPA binds the single-stranded DNA overhangs and activates ATR/ATRIP, Claspin and CHK1, thereby initiating HR [7]. RPA is then replaced by RAD51 with the involvement of several HR proteins [5]. One-ended DSBs arise when a replication fork collides with a single-strand DNA break (SSB) or when a replication fork collapses [8]. They are repaired by break-induced replication (BIR). Stalled replication forks are usually protected from nucleolytic degradation by numerous HR factors [9] or SMARCAL1 and ZRANB3 [10]. However, insufficient stabilization leads to collapse or breakage of the replication fork. Cellular replication stress, dNTP depletion, or collision of replication forks with DNA lesions results in fork reversal or fork regression and chicken foot formation [11]. SMARCAL1 is recruited by RPA-bound ssDNA to the replication fork [12], while Poly(ADP-Ribose)-Polymerase 1 (PARP1) is involved in fork reversal [11]. The RPA-bound ssDNA segments at the paused replication fork lead to the activation of ATR, which phosphorylates CHK1. ATR is the most important DNA damage-related kinase in S-phase, where it has a major impact on DNA repair and the regulation of DNA replication [13]. ATR phosphorylates the intra-S phase kinase CHK1, whereupon CHK1 translocates into the nucleus and undergoes autophosphorylation [14]. Under replication stress, CHK1 is associated with phosphorylated Claspin, which enhances ATR-dependent phosphorylation and supports recruitment of CHK1 to the replication fork [15]. Thus, ATR-mediated phosphorylation of CHK1 activates DNA repair and the intra-S phase checkpoint. CHK1 regulates elongation [16] and activation of replication origins, stabilizes replication forks [17], and delays S-phase progression [14,18]. The stalling of replication forks at DNA lesions can be avoided by CHK1-mediated activation of translesion synthesis (TLS) [19]. Besides the stabilization of replication forks, CHK1 is essential for the activation of DNA repair by HR through phosphorylation of RAD51 and BRCA2 [7]. 

It is unclear to what extent the activation of CHK1 influences the sensitivity of HR deficient tumors. Previous methods to identify these tumors focused on the HR defects that result from a BRCA1/2 mutation and a high HR deficiency score (HRD-score). For both a significantly better response to platinum-based chemotherapy was observed [20,21,22,23,24]. In addition to genetic analysis, it is also possible to characterize HR-deficient tumors functionally. The ex vivo tissue slice culture assay analyzes primary tumor samples for their ability to perform HR. This is assessed by the formation of RAD51 foci after DNA damage, where absence of foci formation indicates HR deficiency [25,26]. It remains unclear in both approaches: (i) whether the observed genetic alteration leads to a functional repair defect, (ii) whether the loss of the RAD51 foci formation provides sufficient information about the functionality of HR, and (iii) how overexpression of RAD51 in interaction with CHK1, in its multiple functions, impacts on these processes. This study investigated the latter point, especially with regard to replication stress, in sporadic non-BRCA1/2-mutated TNBC cells.

## 2. Materials and Methods

### 2.1. Clinical in Silico Analysis

Clinical- and mRNA expression data were extracted from the TCGA database from the cBioportal data (http://www.cbioportal.org). For each tumor, the CIN70 score was calculated according to Birbak et al. [3] by adding the expression values of all CIN70 genes from 1400 patients. For the calculation of disease-specific survival (DSS), the survival data of the patients according to CIN70 score, CHK1 or RAD51 mRNA expression were used and the extreme quartiles were plotted and analyzed using a log-rank test. The mRNA expression of RAD51 and CHK1 of the extreme CIN70 score quartiles were plotted depending on the molecular subtype.

### 2.2. Cell Culture and Treatments

All cell lines used in the study were either purchased from the American Type Culture Collection (ATCC, Manassas, VA, USA) or kindly provided by Prof. Dr. H. Wikman. The cell lines were cultivated in DMEM medium with 10% FCS, 2% glutamine, and 1% penicillin streptomycin in incubators at 37 °C, 5% CO_2_ atmosphere and 100% humidity in cell culture flasks. To inhibit PARP1, olaparib was used in increasing concentrations up to 50 µM and incubated for 5 days. For the treatment with mitomycin C (MMC) concentrations up to 1.5 µg/mL for maximum 1 h were used. The inhibition of CHK1 was achieved by using the small molecule inhibitor MK8776 at 2 µM for 2 h.

### 2.3. Homologous Recombination Assay

HR frequency was measured by stable or transient transfection of I-Sce-1-linearized pDR-GFP (Addgene #26475) and DR-oriP-GFP (kindly provided by M. Jasin) plasmids. Briefly, 1 µg linearized plasmid (pDR-GFP) or 0.5 µg (DR-oriP-GFP) linearized plasmid plus 0.5 µg MSCV-N-EBNA1 (Addgene #37954) was transfected into cells using FuGENE (Roche, Basel, Switzerland) in a 1:3 µg/µL ratio according to the manufacturer’s instruction. After 24 h cells were harvested, and the fraction of GFP-positive cells was determined by flow cytometry.

### 2.4. Western Blot and Immunostaining

Total protein was extracted from exponentially growing cells and 40 µg/ml were resolved by SDS-PAGE using a 4%–15% gradient gel (Bio-Rad Laboratories). After transfer and blocking overnight at 4 °C in Odyssey Blocking Buffer (Li-Cor, Lincoln, NE, USA) proteins were detected by primary antibodies against BRCA2 [2A-9] (1:500, kindly provided by Stephen Smith, Leibnitz Institute, Jena, Germany), FANCD2 [FI17] (Santa Cruz, 1:2000), ATR [N-19] (Santa Cruz, 1:1000), CHK1 [2G1D5] (Cell Signaling, 1:750), RAD51 [14B4] (1:2.000, GeneTex, Irvine, CA, USA), PARP1 [C210] (BD, 1:1000), RPA [9H8] (Santa Cruz, 1:1000), pCHK1 [Ser296] (Cell Signaling, 1:1000), pATR [Ser428] (Cell Signaling, 1:1000), pRPA [S4/S8], (Bethyl, Montgomery, TX, USA, 1:1000), β-actin [AC-74] (1:50.000, Sigma, St. Louis, MO, USA) or HSC70 [B6] (Santa Cruz, 1:1000). Primary antibodies were detected with IRDYE 680 conjugated anti-mouse IgG, IRDYE 800 conjugated anti-rabbit IgG (Licor, 1:7500), IRDYE 680 conjugated anti-rabbit IgG (Licor, 1:7.500 or 15.000) or IRDYE 800 conjugated anti-mouse IgG (Licor 1:7.500 or 15.000). For immunofluorescence staining, cells were seeded on culture slides. After treatment cells were fixed, permeabilized and blocked overnight in 3%BSA. Foci were detected using primary antibodies against RAD51 [AB-1] (Calbiochem, 1:500), yH2AX [Ser139] (Millipore, Burlington, MA, USA, 1:250) or RPA [MA34] (Santa Cruz, 1:400), followed by secondary antibodies Alexa Fluor 488 goat anti rabbit IgG (Cell signaling, 1:600), Alexa Fluor 488 goat anti mouse IgG (Cell signaling, 1:600) or Alexa Fluor 594 goat anti-mouse IgG (Cell signaling, 1:500). EdU was stained with Alexa Fluor Azide 594 or 647(Life Technologies, Carlsbad, CA, USA, 1:500), nuclei were stained with DAPI and the samples were mounted (Vector Laboratories). Fluorescence images were captured using a Zeiss Axioplan 2 epifluorescence microscope equipped with a charge-coupled device camera and Axiovision software. For quantitative analysis, foci were counted by fluorescence microscopy using a 1000-fold magnification. There were 100 cells per dose per slide and experiment were evaluated blindly.

### 2.5. Transmission Electron Microscopy

Treated cells were fixed with 2% paraformaldehyde and 0.05% glutaraldehyde in PBS. Fixed samples were dehydrated using increasing concentrations of ethanol and infiltrated with LR White resin overnight (Plano, Wetzlar, Germany). Subsequently, samples were embedded in fresh resin with accelerator at 37 °C until the resin was polymerized. Ultrathin sections (70 nm) were cut on a Microtome Ultracut UCT (Leica, Wetzlar, Germany) with diamond knives (Diatome, Biel, Switzerland), gathered up on pioloform-coated nickel grids and processed for immunogold-labeling. To block nonspecific staining sections were placed on drops of blocking solution (Aurion, Wageningen, The Netherlands). Afterwards sections were rinsed and incubated with primary antibodies against yH2AX (Millipore, 1:250) and RPA (Santa Cruz, 1:400). After rinsing, secondary antibodies conjugated with 6-nm or 10-nm gold particles (Aurion, Wageningen, The Netherlands) were applied to the grids for 1.5 h. Sections were then rinsed and fixed with 2% glutaraldehyde in PBS. All sections were stained with 3% uranyl acetate and examined using Tecnai BiotwinTM transmission electron microscope (FEI, Eindhoven, The Netherlands). Detection, localization and counting of gold beads and clusters were performed at the electron microscope by eye.

### 2.6. DNA Fiber Assay

Exponentially growing cells were pulse labeled with 25 μM CldU (Sigma) followed by 250 μM IdU (Sigma) for 30 min each. HU was given for 2 h between both labels, Inhibitors 2 h before labelling, MMC was given at the beginning of the 2nd pulse labelling time. Labeled cells were harvested, DNA fiber spreads prepared and stained as described [27]. Fibers were examined using an Axioplan 2 fluorescence microscope (Zeiss, Oberkochen, Germany). CldU and IdU tracks were measured using ImageJ (version 1.48, Company, City, State abbrev. If USA, Country) [27,28]. At least 300 forks were analyzed.

### 2.7. Clonogenic Survival

For survival assays 250 cells were seeded in a 6-well plate 6 h before treatment and cells were cultured for 14 days. Cells were fixed and stained with 1% crystal violet (Sigma-Aldrich, St. Louis, MO, USA). Colonies with more than 50 cells were determined microscopically and normalized to untreated samples. Each survival curve represents the mean of at least three independent experiments.

### 2.8. Statistical Analysis

Statistical analysis, curve fitting and graphs were performed using Prism 6.02 (GraphPad Software, San Diego, CA, USA). Data are given as mean (+SEM) of 3–5 replicate experiments. Unless stated otherwise, significance was tested by Student’s *t*-test.

## 3. Results

### 3.1. Long-Term Disease Specific Survival in Luminal and Triple-Negative Breast Cancer Tumors Depends On RAD51 and CHK1 mRNA Expression

The relationship between CIN and prognosis has already been investigated in a large number of tumors, whereby a high CIN70 value was often associated with a worse prognosis [3]. However, it is unclear whether CIN70 affects both good and poorly treatable subgroups of breast cancer. To clarify this, the effect of the CIN70 score on disease-specific survival (DSS) was analyzed in all subgroups, as well as only in LumA and TNBC using the two extreme quartiles (Figure 1A–C and Appendix A). Irrespective of the subtype, tumors with high CIN70 showed significantly worse 5- and 10-year DSS compared to patients with tumors harboring a low CIN70.

The cause for an increased CIN70 could be a defect in the DNA repair pathway homologous recombination or the DNA damage response. To test this, the role of mRNA expression of RAD51 and CHK1 in respect to CIN70 was analyzed. It was noticeable that the RAD51 expression is significantly higher in CIN high (Figure 1D and Appendix A) than in CIN low tumors. This effect was especially pronounced in the TNBC subtype. A significantly increased expression in CIN high compared to CIN low was also found for CHK1 (Figure 1E and Appendix A) in both subtypes. Consequently, increased expression of RAD51 and CHK1 in CIN high tumors had a negative effect on survival (Figure 1F,G). Patients whose tumor had a high expression of RAD51 showed significantly lower DSS after both 5 and 10 years compared to low RAD51 expression. For the expression of CHK1, the negative effect of a high expression on survival was even more evident. These data suggest that both increased expression of RAD51 and CHK1 may lead to a decreased DNA repair or DNA damage response resulting in tumors with increased chromosomal instability [27].

### 3.2. No Correlation of HR Capacity, Replication Fork Protection, and Sensitivity to PARP1 Inhibition and MMC Treatment

Numerous publications have shown that contrary to the assumption that the more protein present, the more DNA repair capacity can be expected—high expression of RAD51 does not lead to improved HR capacity [27,28,29,30,31]. However, the effect of a high CHK1 expression in tumors has not yet been investigated. It is possible that differences in genetic background or individual cellular adaptation strategies such as tolerance to replication stress are responsible for this phenomenon. To test these aspects, a genetically related system of TNBC was chosen consisting of three MDA-MB-231-derived cell lines, which differ only in their metastatic pattern. While MDA-MB-231 metastasizes to all organs, BR is exclusively colonized in the brain [32] and SA in the bone marrow [33]. For comparison, the luminal cell line MCF7 was used. With this system we first investigated the influence of the expression of DNA repair proteins on the various functions of HR, such as double-strand break (DSB) repair, replication fork protection and cellular survival after treatment with PARP1 inhibitor or MMC. A comparable upregulation of CHK1 and RAD51 on the protein level was detected in TNBC cell lines, showing a TNBC-associable expression pattern (Figure 2A, Appendix A). The MDA-MB-231 showed a slightly increased expression of CHK1 and a further increase in BR and SA compared to the luminal cell line MCF7. The same pattern is observed for the expression of RAD51 with a significant increase in MDA-MB-231, BR, and SA compared to MCF7 cells. Differences in HR based on a higher S/G2 phase fraction can be excluded. (Appendix A).

Due to the different DNA structure represented by a DSB somewhere in the genome and a DSB adjacent to an active replication fork, we used two DNA repair constructs that allow both situations to be simulated. For this purpose, two repair constructs were used to investigate the DNA repair capacity by HR of a single DNA double-strand break (DR-GFP) (Figure 2B,C) or a DNA double-strand break adjacent to an origin of replication (DR-ori-GFP) (Figure 2D), transfected and analyzed by flow cytometry [34]. Significantly lower HR capacities for frank DSBs and DSBs adjacent to a DNA replication origin were found in all TNBCs compared to MCF7 cells. It is striking that the original cell line had the lowest HR capacity in both constructs, BR a slightly higher and SA the highest HR capacity. Thus, increased expression of RAD51 and CHK1 is not associated with increased DNA DSB repair capacity in TNBC.

In addition to DSB repair, HR proteins are required for the stabilization of replication forks, protecting them from nucleolytic degradation, with RAD51 and CHK1 being critical factors [9,35]. To test their protective function, the DNA fiber assay was performed after depletion of the nucleotide pool by addition of hydroxyurea (HU) (Figure 2E). Although all cell lines showed a shortening of the CldU tract, MCF7 with the highest HR capacity showed the most pronounced shortening, followed by MDA-MB-231 and BR. SA showed the strongest protection and only a minimal shortening of the CldU tract.

Next, we tested whether the difference in HR capacity is reflected by sensitivity to PARP1 inhibition by olaparib or MMC treatment in the colony formation assay. Figure 2F shows cellular survival after increasing concentrations of olaparib (left) and MMC (right) for the four cell lines studied. Although the HR capacity showed clear differences, this was not reflected by the sensitivity against PARP1 inhibition since all cell lines showed comparable IC50 values between 1.9 ± 0.2 µM for the cell lines with the highest and 8.5 ± 0.2 µM for the cell line with the lowest HR capacity. The same effect, but to a greater extent, was observed after treatment with MMC. Again, the cell line with the lowest HR capacity showed a 10-fold higher resistance. Thus, there was no correlation between HR capacity, and cellular sensitivity to chemotherapeutic agents exhibiting their toxic effect at the replication fork.

### 3.3. MMC Sensitivity Is Associated with DNA Damage Foci Formation in the S-Phase

Functional HR is characterized by the formation of RAD51 foci after damage followed by their removal after successful DNA repair in S- and G2-phase. Figure 3A shows the formation of RAD51 foci after treatment with MMC in pulse-labeled EdU-positive cells. All cell lines were able to form RAD51 foci; however, MMC-resistant TNBCs showed only a weak induction compared to the sensitive lines. A comparable pattern was also observed for the number of γH2AX foci, with a significantly stronger fold increase in sensitive compared to resistant cell lines. These results indicate that HR is the preferred DNA repair pathway in the two resistant cell lines in the S-phase.

In order to analyze whether the high amounts of RAD51 and γH2AX-foci in the MMC-sensitive cells arise from increased replication stress RPA-foci after MMC treatment were quantified in EdU-positive cells (Figure 3B). The two sensitive cell lines clearly showed 1.5 to 2 times higher amounts of RPA foci than the resistant cell lines without exogenous damage. This increased replication stress in the sensitive cell lines also occurred after treatment with MMC, with on average significantly more RPA foci compared to the resistant cell lines.

To further localize the occurrence of DNA damage in the S-phase, replication-associated DSBs were visualized by electron microscopy by parallel labeling of γH2AX and RPA after mitomycin C in one of the sensitive and resistant cell lines. (Figure 3C, Appendix A). In the sensitive cell line, an accumulation of γH2AX adjacent to isolated RPA foci in heterochromatic areas (dark staining) was already observed in the untreated state, while the resistant line showed both RPA and γH2AX rather sporadically and broadly distributed and less frequent.

This trend intensified further after treatment with MMC and reveals the accumulation of several γH2AX signals in the sensitive cell line around a single RPA signal, whereas in the resistant cell line both markers remained scattered in the nucleus (Appendix A). This suggests that the sensitive cell line shows increased DSBs at stalled replication forks, generally also referred to as replication stress. The data very clearly indicate that MMC sensitivity is most likely due to increased replication stress in the two sensitive cell lines.

### 3.4. Activation of DNA Damage Response Leads to Resistance to MMC by Avoiding Replication Stress.

The potential mechanism underlying the differential cellular sensitivity to MMC was investigated by visualization of replication processes [36]. Figure 4A shows the frequency distribution of DNA strand lengths, i.e., before actual damage by MMC or after MMC treatment (Appendix A), compared to untreated control. The sensitive cell lines MCF7 and BR showed significantly shorter DNA strands after damage with MMC than in the untreated control, while the two resistant cell lines showed moderate or no shortening. These observations demonstrate that the differential HR capacity shown in Figure 2B–D cannot explain the observed differences in cellular sensitivity to MMC, which instead appear to be linked to the protection of replication tracts, and therefore to the DNA damage response at the replication fork. To verify this, the activation of ATR, CHK1, and RPA was investigated.

Figure 4B shows the phosphorylation of ATR, CHK1, and RPA after MMC (Appendix A). The phosphorylation of CHK1 was clearly associated with resistance to MMC, and showed a more than two-fold activation in the two resistant cell lines compared to the two sensitive. This activation, which can be clearly assigned to resistance, is also reflected in the reduced phosphorylation of RPA showing a 7.0 ± 0.4 and 6.0 ± 0.02 fold increase in the two sensitive cell lines but not in the two resistant ones. The activation of ATR, on the other hand, showed no consistent response in the sensitive and resistant cell linesThese activation patterns suggest that CHK1 plays a crucial role in MMC resistance. To test this hypothesis, we treated the cells with MMC in the presence of a CHK1 inhibitor and determined the length of the DNA strands with the DNA fiber assay. Figure 4C shows the frequency distribution of CldU or IdU (Appendix A) labeled DNA fibers, i.e., before MMC damage, but in the presence of the CHK1 inhibitor MK8776. As expected, the two resistant cell lines showed a significant effect in response to the CHK1 inhibitor, while there were only minor effects present in the sensitive ones. This confirms the assumption that the observed resistance is due to an increased activation of CHK1.

### 3.5. Activation of CHK1 Protects against DNA Damage in the S-Phase and Mediates Resistance to MMC in HR-Deficient Cell Lines

Next, it was investigated whether the observed replication stress after inhibition of CHK1 in the two resistant cell lines had a more pronounced effect on DNA damage and cellular survival. Figure 5A shows the occurrence of a pan-nuclear γH2AX signal as a marker of the replicative catastrophe [37] after MMC damage. While in the two sensitive cell lines a large proportion of cells with pan-nuclear γH2AX signal could be observed following MMC treatment, the two resistant lines showed a 3–4 times lower percentage of pan-nuclear γH2AX positive cells. Conversely, after combined treatment with MMC and CHK1 inhibitor, the percentage of pan-nuclear γH2AX positive cells increased 6- and 2.5-fold respectively in the two resistant cell lines, whereas the two sensitive lines showed only a 1.2-fold increase relative to MMC mono treatment. These results suggest that MMC treatment alone was already sufficient to induce the replicative catastrophe in sensitive cells. In the resistant cells, this effect was only achieved to a comparable extent when MMC was combined with CHK1 inhibition.

## 4. Discussion

In this study, we showed a negative association of a high CIN70 score with DSS in breast cancer and confirmed data from previous studies [2,3]. Furthermore, we observed that a high CIN 70 score determines survival in both good and poorly curable subgroups of breast cancer. The high CIN70 score in LumA corresponds to the low CIN70 score of the TNBC. This implies that the use of a more aggressive therapy for LumA with a high CIN70 score could possibly lead to significantly higher survival rates. An increased DNA repair capacity to compensate the higher CIN with a resulting therapy resistance as the cause of the poor prognosis for breast as well as other tumors [3]. We confirm here that this assumption applies not only to TNBCs but also to LumA, with a significantly increased mRNA expression of RAD51 and CHK1 in tumors with high CIN scores. Accordingly, DSS was negatively associated with increased expression of both genes. This confirms data from us and others showing an association of high expression and poor prognosis for RAD51 [38,39,40,41], as well as CHK1 [41,42,43] at the protein and mRNA levels, respectively. However, there is also a highly significant increase in mRNA expression with CIN, both for RAD51 and even stronger for CHK1. This observation clearly shows that more DNA repair protein does not lead to increased DNA repair. It seems more likely that the superior DNA damage response with cell cycle checkpoints or increased tolerance to DNA damage is causal. Both options would lead to decreased response and poorer survival of treated patients. In the present study, the general DNA damage response appears to be the decisive factor for the observed therapy resistance in TNBCs.

In addition, we showed that TNBC cells display no degradation of replication forks and no increased olaparib sensitivity despite a reduced HR capacity. Rather, resistance was observed after MMC with adequate RAD51 focus formation and correspondingly low levels of the DNA damage marker γH2AX in the S-phase. This was due to a pronounced activation of the DNA damage response by CHK1, which ensured unimpaired replication and could be reversed by CHK1 inhibition. Thus, this study demonstrates an important role of CHK1 which can compensate for a reduced HR capacity by preventing replication stress in TNBC.

A correlation of increased RAD51 and CHK1 expression was observed in several studies and was significantly associated with TNBC status [44]. The low HR capacity in TNBC cells observed here also confirms the described limited HR functionality in TNBCs in the clinical setting [21]. Despite high RAD51 expression and functional BRCA1 a low HR capacity was observed by the plasmid reconstruction assay. This could be attributed to a lack of activation of CHK2 via ATM [45]. It is more likely that the induction of a single DSB activates neither ATM, nor ATR [8]. The limited validity of the detection of a single DSB is strongly supported by the observation that cells without BRCA1 expression show a high HR capacity for a replication-independent DSB and virtually no HR capacity for a replication-associated DSB (Appendix A).

One of the further functions of HR is the protection of replication forks. Cells with the highest HR capacity showed the highest instability of replication forks. Reduced protection of replication forks has already been observed in tumor cells despite functional HR [9]. This distinguishes tumor from non-tumor cells for which stabilization of replication forks by HR factors is essential [46].

Despite their low HR capacity, MDA-MB-231 were resistant to olaparib and MMC. Both agents lead to one-ended DSB, which occur when a replication fork hits a single strand break and collapses [47,48]. A replication run-off occurs, the replication machinery dissociates and a DNA end is released [49] which is the substrate for the third function of HR [47]. The one-ended DSB resulting from MMC treatment represents a more complex repair situation for the cell and activates both nucleotide excision and Fanconi repair factors [50]. Only after successful activation of FANCD2 a substrate for HR is provided in the further repair process [51,52]. All cell lines could successfully form RAD51 foci according to their cellular survival after MMC. Thus, no HR defect was observed.

The processes of HR at the replication fork, such as fork protection and repair of one-ended DSB, are controlled by the intra-S phase damage response kinase ATR and its downstream kinase CHK1 [45]. The replication fork stabilizing function of CHK1 can be observed after PARP1 inhibition and MMC treatment and is independent of ATR, RAD51, and BRCA1 [35,45,53,54]. The resistance mechanism may involve avoidance of one-ended DBSs or activation of alternative DNA repair pathways [14,55]. More obvious is the direct negative effect of CHK1. The inhibition of CHK1 leads to the increased initiation of replication and increased phosphorylation of further ATR targets. This results in production of more DSBs, reduced MRE11 activity [17] and an affected HR function due to the phosphorylation of RAD51 [56]. Thus, efficient utilization of the ATR-CHK1 signaling cascade can compensate for reduced HR function. This is confirmed by studies in ovarian carcinoma [57]. Thus, the activation of CHK1 could influence the sensitivity of HR deficient tumors. Therefore, the established methods to successfully identify HR deficient tumors should be extended by the detection of CHK1 activation. This could be made possible by extending BRCA1/2 testing by genes of the intra S-phase damage response or the combined detection of RAD51 with RPA in S-phase cells [25,26,58].

## Figures and Tables

**Figure 1 cells-09-00238-f001:**
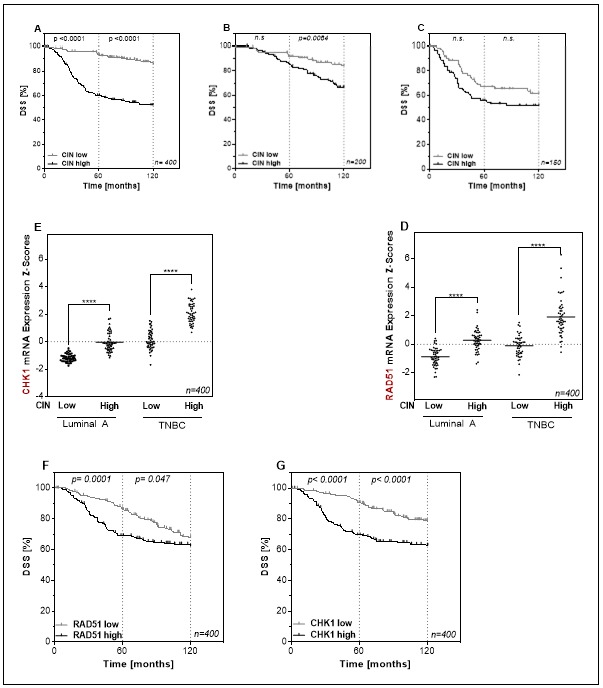
A high gene expression profile consisting of 70 genes associated with a functional aneuploidy (CIN70) score is negatively associated with disease-specific survival in luminal A (LumA) and triple-negative breast cancer (TNBC) subtypes and correlates with the expression of RAD51 and CHK1. (**A**–**C**). Kaplan–Meier analysis of the CIN70 score as prognostic factor for disease-specific survival (DSS) in breast cancer patients (**A**) (*n* = 400) patients with LumA (**B**) (*n* = 200) and TNBCs (**C**) (*n* = 150) using the two extreme quartiles. The CIN70 score defines the differential mRNA expression of 70 genes in tumors classified as stable and unstable based on their structural and numerical chromosomal alterations [3]. DSS is plotted against time after therapy. All, LumA and TNBC tumors with CIN high showed significantly worse 5- and 10-year DSS compared to tumors with CIN low, with 52% vs. 86% after 10 years for all, 84% vs. 67% after 10 years (n.s/*p* = 0.0084) for LumA and 52% vs. 62% after 10 years (n.s/n.s) for TNBC tumors. *p*-values were calculated on the basis of the log-rank test. (**D**,**E**). mRNA expression of RAD51 (**D**) and CHK1 (**E**) for LumA and TNBC tumors of the Metabric data set sorted by CIN70 score using the two extreme quartiles. RAD51 is expressed significantly higher in CIN high than in CIN low tumors, both in LumA, with −0.88 ± 0.09 vs. 0.27 ± 0.1 and −0.11 ± 0.01 vs. 1.92 ± 0.19, respectively. A significantly increased expression in CIN high compared to CIN low was also found for CHK1, with 1.17 ± 0.04 and −0.03 ± 0.096 for LumA and 0.13 ± 0.1 vs. 2.08 ± 0.1 for TNBC. (**F**,**G**). Kaplan–Meier analysis of DSS of 400 patients according to the RAD51 (**F**) or CHK1 (**G**) expression using the two extreme quartiles. DSS is plotted against time after therapy. Patients whose tumor had a high expression of RAD51 showed significantly lower DSS compared to low RAD51 expression (68% vs. 62%). For the expression of CHK1, the negative effect of a high expression on survival was even more evident (62% vs. 80%). *p*-values were calculated on the basis of the log-rank test (**** *p* < 0.0001; Student’s *t*-test).

**Figure 2 cells-09-00238-f002:**
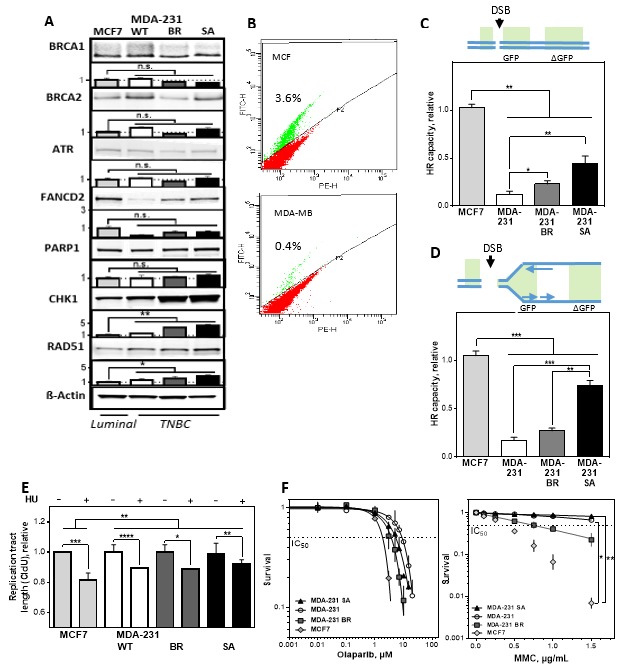
Low homologous recombination (HR) capacity is compatible with efficient replication fork protection and resistance to PARP1 inhibition and mitomycin C (MMC) in TNBCs. (**A**) Immunoblot detection of BRCA1, BRCA2, ATR, FANCD2, PARP1, CHK1 and RAD51 derived from total cell extracts of exponentially growing MCF7 and MDA-MB-231/BR/SA cells. The MDA-MB-231 showed a slightly increased expression of CHK1 (1.49 ± 0.1) and a further increase in BR (3.7 ± 0.1) and SA (4.3 ± 0.3) compared to MCF7 cells. The same pattern is observed for the expression of RAD51 with a significant increase of 1.64 ± 0.3 in MDA-MB-231, 2.4 ± 0.5 and 2.6 ± 0.3 in BR and SA compared to MCF7 cells. ß-Actin was used as the loading control. Data are presented as mean ± SEM. Immunoblot signals were detected and quantified by a LiCor system. (**B**–**D**) Repair of open and replication-associated DNA double-strand break (DSB) as a measure of HR capacity, determined by plasmid-reconstruction assay and analyzed by FACS. Cells were transiently transfected with the pDRGFP construct (**C**) or DR-ori-GFP plus the ori-activating MSCV-N EBNA1 construct (**D**) for 24 h. The number of GFP-expressing cells was normalized to the absolute HR capacity of MCF7. A significantly lower HR capacity for frank DSB was found in all TNBCs compared to MCF7 cells, with 0.12 ± 0.008 for MDA-MB-231, 0.23 ± 0.04 for BR, and 0.45 ± 0.07 for SA. The same pattern was observed for DSBs adjacent to a DNA replication origin, with 0.17 ± 0.03 for MDA-MB-231, 0.27 ± 0.03 for BR, and 0.75 ± 0.05 for SA compared to MCF7 cells. (**E**) Mean length of DNA fibers in MCF7 and MDA-MB-231/BR/SA cells. The cells were sequentially labelled with CldU and IdU for 30 min and treated with HU between both labels for 4 h. DNA was spread on slides, fixed and incorporated nucleotides were detected by immunofluorescence. Although all cell lines showed a shortening of the CldU tract, MCF7 showed the most pronounced shortening with 0.8 ± 0.004, followed by MDA-MB-231 and BR with 0.89 ± 0.003 and 0.89 ± 0.001. SA showed only a minimal shortening of the CldU tract with 0.93 ± 0.006. The length of the DNA fibers was measured with the Image J software and calculated relative to the absolute length of the untreated controls. (**F**) Cellular survival after treatment with olaparib (left) or MMC (right) in MCF7 and MDA-MB-231/BR/SA cells. The cells were seeded 24-h prior treatment with olaparib or MMC for 5 days or 1 h, fixed after 14 days, and the number of colonies was counted. Shown are means from three independent experiments ± SEM. Asterisks represent significant differences (* *p* < 0.05; ** *p* < 0.01; *** *p* < 0.001; **** *p* < 0.0001, n.s. not significant; Student’s *t*-test).

**Figure 3 cells-09-00238-f003:**
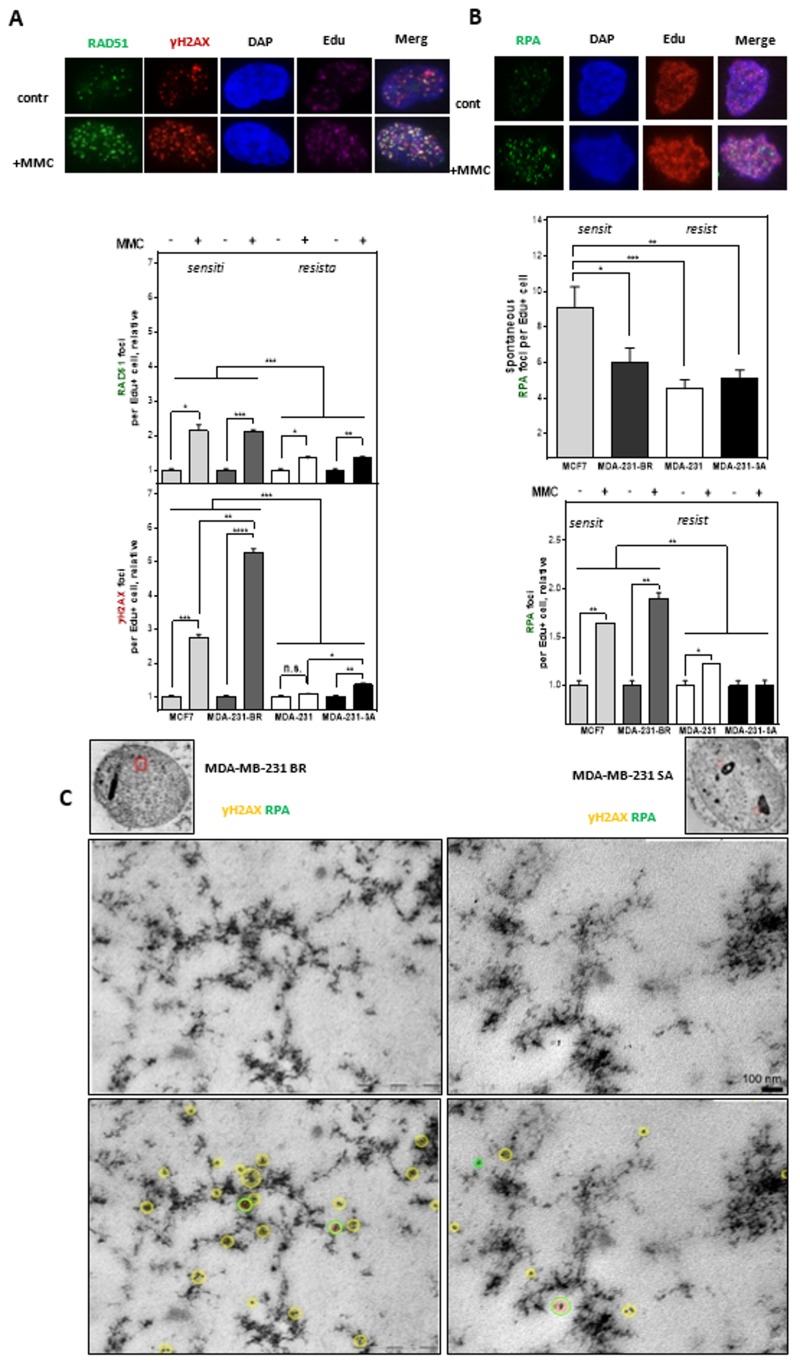
Resistance to MMC is due to the improved repair of DNA damage at DNA replication forks. (**A**) RAD51(green) and γH2AX foci (red) as well as (**B**) RPA foci (green) in S-Phase (Edu+) cells arising spontaneously or after treatment with MMC. Cells were treated with 0.5 µg/mL MMC for 1 h after pulse labeling with 10 µM Edu for 20 min. Immunofluorescent staining was performed 24 h after treatment with γH2AX or RPA and fluorescent second antibodies. Replicating cells were discriminated by incorporated Edu stained with the “click-it” reaction. Foci analysis was done with the Image J Software. Foci were only counted in Edu-positive nuclei (*n* = 100). DNA was counterstained by DAPI. The number of Foci was calculated relatively to the number of Foci in untreated control. Shown are means of three independent experiments ± SEM. Asterisks represent significant differences (* *p* < 0.05; ** *p* < 0.01; *** *p* < 0.001; **** *p* < 0.0001, n.s. not significant; Student’s t-test). (**C**) Transmission electron microscopy shows colocalization of gold-labeled γH2AX (yellow) and RPA (green) for MDA-MB-231 BR and MDA-MB-231 SA cells in untreated and MMC treated cells (0.5 µg/mL) 24 h after treatment within nuclear ultrastructure mainly associated to heterochromatic regions.

**Figure 4 cells-09-00238-f004:**
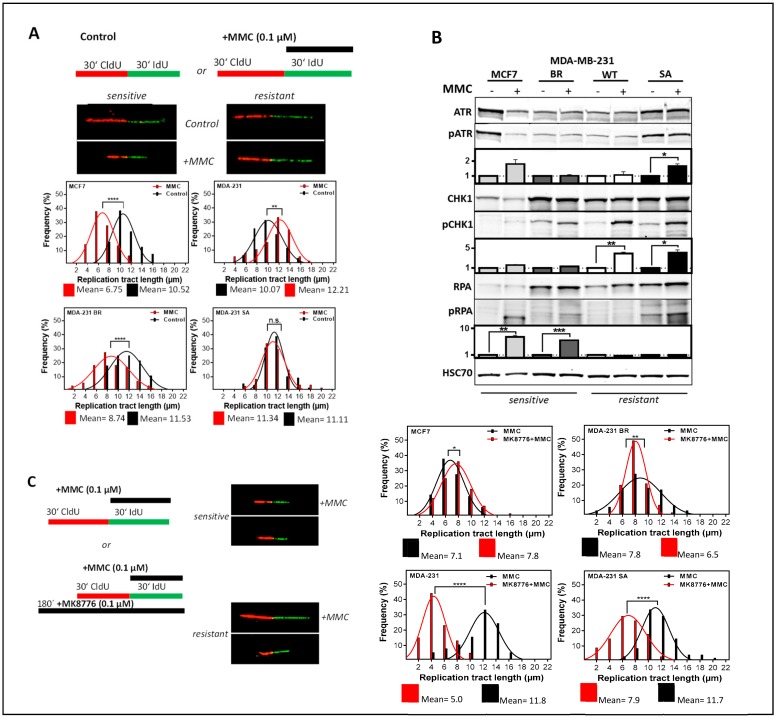
CHK1 inhibition leads to increased replication stress only in MMC-resistant TNBC. *(***A**) Examples and frequency distribution of DNA fiber lengths (CldU) in untreated and MMC treated cells. Exponentially growing cells were sequentially labeled with CldU and IdU in the absence or presence of MMC (0.1 µM). DNA was spread and incorporated CldU and IdU was detected with appropriate antibodies. Shown are means ± SEM of DNA fiber length (CldU) frequency distributions of three independent experiments. Asterisks represent significant differences (** *p* < 0.01; *** *p* < 0.0001, Student’s *t*-test). (**B**) Immunodetection of activated intra-S phase checkpoint proteins. Cells were treated with 1.5 µg/mL MMC for 1 h and proteins were extracted 24 h later. Proteins were separated and transferred by Western blotting. Detection of proteins was performed with appropriate antibodies. HSC70 served as the loading control. Phosphorylation of the untreated control was used for standardization and ratios of phosphorylated to non-phosphorylated protein are shown. Data from three independent experiments were used for quantification. Errors are mean values + SEM. (**C**) DNA fiber lengths of CldU labeled tracts after treatment with MMC in the presence of the CHK1 inhibitor MK8776 (1 µM). Exponentially growing cells were incubated for 2 h with MK8776 and sequentially labeled with CldU and IdU (plus 0.1 µM MMC), DNA was spread and incorporated nucleotides were detected with the appropriate antibodies. The frequency distribution of DNA fiber lengths in the first label (CldU) of three independent experiments is shown. Asterisks represent significant differences (* *p* < 0.05; ** *p* < 0.01; *** *p* < 0.001; **** *p* < 0.0001, n.s. not significant; Student’s t-test).

**Figure 5 cells-09-00238-f005:**
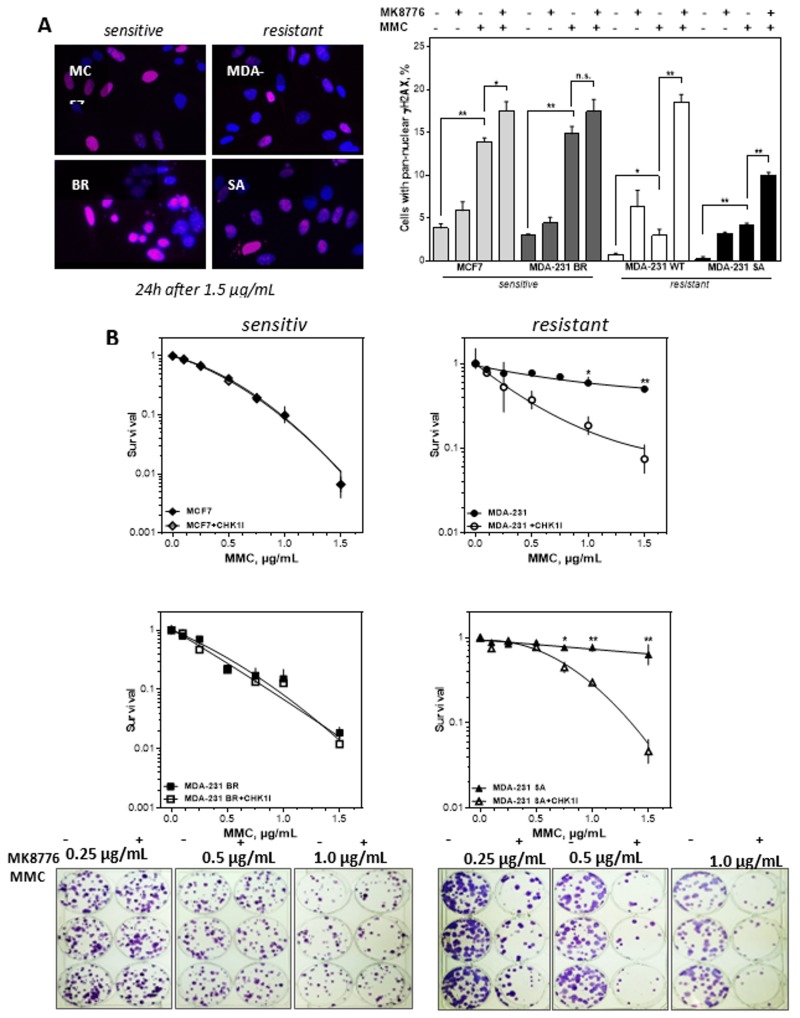
CHK1 inhibition sensitizes only cells with resistance towards MMC. (**A**) Percentage and representative examples of cells with pan-nuclear yH2AX signal (red) after treatment with MMC alone or combined with CHK1 (MK8776) inhibition. Exponentially growing cells were treated with 0.5 µg MMC for 1 h after being incubated with 2 µM MK8776 for 2 h. The 24 h after treatment immunofluorescence staining for γH2AX was performed and nuclei were counterstained with DAPI. Cells with pan-nuclear γH2AX signal indicating replication stress were microscopically evaluated. Means + SEM of three independent experiments are shown. Asterisks represent significant differences (* *p* < 0.05, ** *p* < 0.001, *** *p* < 0.0001, Student’s *t*-test). (**B**) Cellular survival after treatment with MMC alone or in combination with the CHK1 inhibitor MK8776. Cells were plated, treated with MK8776 (2 µM) for 2 h and/or MMC for 1 h, fixed and stained after 14 days and the number of colonies was counted. Adding MK8776 sensitized the two resistant cell lines to MMC (*p* = 0.003 and 0.007 at 1.5µg/mL MMC). For the two MMC sensitive cells there was no sensitizing effect by the CHK1 inhibitor. Shown are means of three independent experiments ± SEM. Asterisks represent significant differences (* *p* < 0.05; ** *p* < 0.01, Student’s *t*-test). Induction of the replicative catastrophe should result in cell death, which was investigated by the colony formation assay. Figure 5B shows that adding the CHK1 inhibitor MK8776 sensitized the two resistant cell lines to MMC. The IC50 values for the two cell lines were reduced by an enhancement factor of 4.3 and 2.9. For the two MMC sensitive cells there was no sensitizing effect by the CHK1 inhibitor. Thus, only cell lines resistant to MMC could be sensitized by inhibition of CHK1 (* *p* < 0.05; ** *p* < 0.01; *** *p* < 0.001; **** *p* < 0.0001, n.s. not significant; Student’s t-test).

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
