# Peer review of "Prevention of DNA Replication Stress by CHK1 Leads to Chemoresistance Despite a DNA Repair Defect in Homologous Recombination in Breast Cancer"

_cells, 2020, doi:10.3390/cells9010238_

Round 1

Reviewer 1 Report

Studies showed that some triple-negative breast cancer is related with a defect in HR-mediated DNA-double-strand-break repair. However, it is unclear how the activation of CHK1 influences the sensitivity of HR deficient tumors. This study investigates the regulation of HR by the intra-S-phase damage response at the replication. The results showed that CHK1 has a preferred role over HR in preventing replication stress in TNBC. The signaling cascade ATR-CHK1 can compensate for a double-strand break repair error and lead to resistance of HR-deficient tumors. The paper’s logic flows well and a lot of data has been collected.

Major points

In figure 1, is there an overlap between the samples from figure 1A and figure 1B/C? There is no significant difference in both LumA and TNBC patients at 5 years, but there is a significant difference in figure 1A. In figure1, what are the criteria for high and low CIN and high and low RAD51/CHK1? In line 193, the authors need more evidence to claim the causal relationship between the increased expression of RAD51/CHK1 and decreased DNA repair even though they are related. Inline 236, please explain the function of DR-GFP and DR-ori-GFP, and how it can detect the HR capacity. In figure 3C, the authors need a way to quantify the localization of H2AX and RPA, then get some statistical results.

Minor points

In line 44, pleases add the full name of HR, homologous recombination, when it first appears. In line 272, please indicate what is HU (hydroxyurea?) and the function of HU. In figure 2E, the significance analysis should be done between HU-positive MCF7 and other groups. In line 395-402, please separate the main content with the figure legend. In figure 3, please correct  “yH2AX” to “gH2AX”. There are no scale bars in Figure 3 A and B, and also in Figure 5 A. Spaces should be deleted in lines 205, 237, 407, and 457. Please delete the extra “E” in line 258.

Author Response

Studies showed that some triple-negative breast cancer is related with a defect in HR-mediated DNA-double-strand-break repair. However, it is unclear how the activation of CHK1 influences the sensitivity of HR deficient tumors. This study investigates the regulation of HR by the intra-S-phase damage response at the replication. The results showed that CHK1 has a preferred role over HR in preventing replication stress in TNBC. The signaling cascade ATR-CHK1 can compensate for a double-strand break repair error and lead to resistance of HR-deficient tumors. The paper’s logic flows well and a lot of data has been collected.

First of all, we would like to thank the reviewer for the very positive evaluation of our manuscript and are pleased to have the chance to work on the open questions and points of criticism and to be able to resubmit the manuscript.

1. In figure 1, is there an overlap between the samples from figure 1A and figure 1B/C?

Yes, there is an overlap between the analyzed samples from figure 1A-C. Figure 1A compares the upper and lower quartiles of all breast cancer subgroups, Figure 1B only the subgroup of luminal A tumors and Figure 1C only triple-negative tumors. This is now more clearly indicated in the revised manuscript by the heading "all subgroups" in Figure 1A.

2. There is no significant difference in both LumA and TNBC patients at 5 years, but there is a significant difference in figure 1A.

Exactly, there is a significant difference up to 5 years when all subgroups are analyzed, while there is no significant difference up to 5 years in the subgroups LumA/TNBC. In the case of TNBCs, the main reason will be that only 150 samples were analyzed. This is because even in the very large MetaBric cohort only about 10% are TNBCs. In the group of luminal A, a significance will only become apparent after 4-8 years. This could be due to the fact that in both quartiles proliferating tumors still dominate, but later proliferation-independent factors play a role.

3. In figure1, what are the criteria for high and low CIN and high and low RAD51/CHK1?

For the determination of high and low CIN the criteria of the CIN70score defined by Birkbak et al. 2013 were used. The CIN70 score defines the different mRNA expression of 70 genes in tumors, which were classified as CIN low and CIN high based on their structural and numerical chromosomal aberrations. We apologize for this inaccuracy and have now inserted an explanatory sentence for the CIN70 score in the legend of Figure 1.

4. In line 193, the authors need more evidence to claim the causal relationship between the increased expression of RAD51/CHK1 and decreased DNA repair even though they are related.

We agree with the reviewer that our conclusion may have been a little too general and have softened the statement somewhat. In addition, we have added a literature reference that shows supportive data from us for RAD51 and motivated us to make this general statement.

5. Inline 236, please explain the function of DR-GFP and DR-ori-GFP, and how it can detect the HR capacity.

We apologize for the incomplete explanation of the DNA repair constructs and have now changed the sentence as follows: For this purpose, two repair constructs were used to investigate the DNA repair capacity by HR of a single DNA double-strand break (DR-GFP) (Fig. 2B,C) or a DNA double-strand break adjacent to an origin of replication (DR-ori-GFP) (Fig. 2D), transfected and analyzed by flow cytometry. Furthermore, we have added an explanatory literature reference.

6. In figure 3C, the authors need a way to quantify the localization of H2AX and RPA, then get some statistical results.

We agree with the reviewer that quantification of the localization of H2AX and RPA in figure 3C is desirable to obtain statistically significant results. Statistically significant data for the differences in the distribution of H2AX and RPA can already be seen in Figures 2A and B. Figure 3C is intended to support these data only exemplarily on an electron microscopic basis.

7. In line 44, pleases add the full name of HR, homologous recombination, when it first appears.

We apologize for this omission and have changed this as follows: Homologous recombination-mediated (HR-mediated).

8. In line 272, please indicate what is HU (hydroxyurea?) and the function of HU.

We have changed the sentence according to the expert's note as follows: To test their protective function, the DNA fiber assay was performed after depletion of the nucleotide pool by addition of hydroxy urea (HU).

9. In figure 2E, the significance analysis should be done between HU-positive MCF7 and other groups.

The significances were added accordingly.

10. In line 395-402, please separate the main content with the figure legend.

Thanks for the note we have implemented.

11. In figure 3, please correct "yH2AX" to "gH2AX".

yH2AX was replaced by gH2AX.

12. There are no scale bars in Figure 3 A and B, and also in Figure 5 A.

We cannot show any scaling because the layered images used in our microscope cannot be scaled. If desired, we could add single images with a scale to the supplements.

13. Spaces should be deleted in lines 205, 237, 407, and 457. Please delete the extra "E" in line 258.

We apologize for the typos and have removed them in the resubmitted manuscript.

Reviewer 2 Report

This paper entitled « Prevention of DNA replication stress leads to chemoresistance despite a DNA repair defect in homologous recombination in breast cancer” is proposed by Felix Meyer et al. and aims at studying the role of homologous recombination (HR) in cell aggressiveness. This paper deals with interesting points that are important for understand chemoresistance and is well built but some modifications are required and some points have to be clarified before considering for publication.

Major points:

A major point is that it is difficult to understand what is really new in the article proposed by these authors. They don’t compare enough to literature and explain why they perform these experiments in regards of previous works.

-Figure 1: not clear whether survival discrimination is maintained inside LumA or inside TNBC. This point seems important to support conclusions of the authors

-Figure 1: a correlation test between CHK1, rad51 and CIN score would be useful to understand whether the modifications are progressive or rather ON/OFF. This should be also discussed.

-Figure 2A and 4B: statistical analysis is lacking. (how many independent experiences are done ?). In sup data, only 1 WB is available. Moreover, not clear why some bands are selected (black squares) and adjacent bands (less intense) not ?

-Figure 3C: quantification is required + statistical analysis. Legend of histograms in Fig4B is lacking (phospho/non phospho ratio? or phosopho/HSC70? or other?).

- More information is required for HR and SA cells ; what is already known about these cells and the subject treated by the authors. All the conclusions are obtained with only the differences between MCF7 and variant of MDA-MB-231 cells. It would be very helpful to support the paper whether some key experiments could be reproduced in independent cell models.

Other points:

-some words are not well defined and this makes sometimes the paper difficult to understand.

(For example, CIN (line 39) is not explained (a short definition arrived line 159); complete names of genes are not supplied ; Hu means hydroxyurea (line 272) ?

-complete references of many products are lacking; this is particularly important for antibodies since suppliers generally propose several antibodies against the same target.

-line 115: how blocking is done?

-line 137: which concentration of uranyl acetate is used?

-not clear how colonies containing more than 50 cells are identified

-line 160: the link between high CIN70 score and poorly treatable tumors should be moderated since immunotherapy is being growing in many anti-cancer therapies and works much better in cancers associated with many mutations (generating neo-epitopes)

-word positions (sensitive/resistant) is not very clear. Please add some lines/arrows.

A final schema could be useful to replace all the points in the context.

Author Response

Studies showed that some triple-negative breast cancer is related with a defect in HR-mediated DNA- This paper entitled « Prevention of DNA replication stress leads to chemoresistance despite a DNA repair defect in homologous recombination in breast cancer” is proposed by Felix Meyer et al. and aims at studying the role of homologous recombination (HR) in cell aggressiveness. This paper deals with interesting points that are important for understand chemoresistance and is well built but some modifications are required and some points have to be clarified before considering for publication.

We would like to thank the reviewer for the positive review of our manuscript and would be pleased if the manuscript could be accepted for publication after considering the requested suggestions for improvement.

A major point is that it is difficult to understand what is really new in the article proposed by these authors. They don’t compare enough to literature and explain why they perform these experiments in regards of previous works.

We apologize for the fact that our new findings may not have been made clear enough, but we are not aware of any publication that deals with the superordinate role of CHK1 via HR with respect to chemoresistance in an isogenic cell system. We have therefore changed our title as follows:

"Prevention of DNA Replication Stress by CHK1 Leads to Chemoresistance Despite a DNA Repair Defect in Homologous Recombination in Breast Cancer"

The reviewer notes that in our manuscript we did not make sufficient use of the existing literature. The publications known to us all deal exclusively with either CHK1 or RAD51, except one, which was published by us and is now included in the text (Parplys and Seelbach et al., 2015).

Figure 1: not clear whether survival discrimination is maintained inside LumA or inside TNBC. This point seems important to support conclusions of the authors

We agree with the reviewer that the comparison of CIN low and CIN high is much more evident in the overall cohort than in the two subgroups. In our opinion, this is primarily due to the significantly larger number of patients. In support of this assumption, we have recombined the two subgroups and added them in Supplementary Fig.2A.

For ourselves, however, it was more surprising that in the LumA subgroup the differences in survival after 10 years were significantly different although lumA tumors are characterized by very high survival rates. In our opinion, this is the first time that in LumA tumors a high CIN leads to an almost comparably poor survival as in TNBC.  In our opinion, it is precisely this observation that is the actual starting point for our investigation, namely which subgroup-overlapping factors could be responsible for the poor survival. This is why MCF7 and MDA-MB-231 BR are summarized as sensitive from Figure 3A onwards.

figure 1: a correlation test between CHK1, rad51 and CIN score would be useful to understand whether the modifications are progressive or rather ON/OFF. This should also be discussed.

We thank the reviewer for this good idea and have added the corresponding analyses in the supplementary S1B-D. The analysis shows a clearly progressive increase for CHK1 with CIN70, which is less pronounced for RAD51.

Figure 2A and 4B: statistical analysis is lacking. (how many independent experiences are done ?). In sup data, only 1 WB is available. Moreover, not clear why some bands are selected (black squares) and adjacent bands (less intense) not ?

We thank the reviewer for his comments and have added the statistics in Fig. 2A and 4B in the revised manuscript. As shown in the legend, three experiments per data point were performed. We have added only one western blot in the appendix to show the quality of the westerns. The black framed bands are those used in the figure. Weaker, unused bands are the same extracts applied with a lower amount of protein to discriminate the best signal (Supplentary Fig.S2A,B).

Figure 3C: quantification is required + statistical analysis.

We agree with the reviewer that quantification of the localization of H2AX and RPA in figure 3C is desirable to obtain statistically significant results. Statistically significant data for the differences in the distribution of H2AX and RPA can already be seen in Figures 2A and B. Figure 3C is intended to support these data only exemplarily on an electron microscopic basis. 

Legend of histograms in Fig4B is lacking (phospho/non phospho ratio? or phosopho/HSC70? or other?).

We thank the reviewer for his attention and have now changed the legend of Fig. 4B.

More information is required for HR and SA cells; what is already known about these cells and the subject treated by the authors. All the conclusions are obtained with only the differences between MCF7 and variant of MDA-MB-231 cells. It would be very helpful to support the paper whether some key experiments could be reproduced in independent cell models.

If the reviewer means the two subcell lines BR and SA, there is very little literature on these two cell lines and none that deals with DNA repair. The cell lines were prepared either by inoculation of MDA-MB-231 into the tail vein followed by isolation of brain metastasis (BR) or by cell culture generated subclones of MDA-MB-231 that metastasize exclusively to bone (SA). We would like to apologize for not naming the two primary citations. The cell line MDA-MB-231 BR was established by Yoneda et al, 2001 and the cell line MDA-MB-231 SA by Pollari et al, 2011 and are now added to the reference list.

We also understand that the reviewer considers it desirable to reproduce the data in an independent system. Due to the enormous experimental effort involved in producing subcell lines from multiple metastatic to organ-specific metastatic cell lines, we would prefer to continue this in another publication and then extend it to other subtypes.

Other points:

some words are not well defined and this makes sometimes the paper difficult to understand.

(For example, CIN (line 39) is not explained (a short definition arrived line 159); complete names of genes are not supplied; Hu means hydroxyurea (line 272) ?

We would like to apologize for this and, in accordance with the comments of the other reviewer, have defined some terms earlier and more precisely. We would like to leave it to the editor to decide whether it is necessary to mention the complete names of all genes mentioned in the manuscript

complete references of many products are lacking; this is particularly important for antibodies since suppliers generally propose several antibodies against the same target.

The corresponding references for the antibodies have now been added.

line 115: how blocking is done?

Overnight blocking at 4°C in Odyssey Blocking Buffer (Li-Cor, Nebraska, USA) which is now mentioned in the MM part.

line 137: which concentration of uranyl acetate is used?

A concentration of 3% uranyl acetate was used.

not clear how colonies containing more than 50 cells are identified

This was done microscopically and is now described in the text as follows: Colonies with more than 50 cells were determined microscopically and normalized to untreated samples.

line 160: the link between high CIN70 score and poorly treatable tumors should be moderated since immunotherapy is being growing in many anti-cancer therapies and works much better in cancers associated with many mutations (generating neo-epitopes).

We agree with the reviewer that tumor therapy is currently undergoing major changes and should be modified for current or future approaches. The data sets used in this study are all retrospective and therefore from the time before immunotherapy was used. In addition, the application of immunotherapy for breast cancer is currently still questionable, as the mutation burden is lower than for other tumors. We would therefore like to ubstain from changing the text according to the wishes of the rewiewer.

word positions (sensitive/resistant) is not very clear. Please add some lines/arrows.

We would like to thank the reviewer for his suggestion, and have changed all labels regarding resistant and sensitive cell lines.

a final schema could be useful to replace all the points in the context.

We agree with the reviewer that a final model always rounds off a data set well. However, we have used a very well defined, but limited cell system, so we prefer to avoid a general model.

Round 2

Reviewer 1 Report

The authors have responded to my comments well. They also made some corresponding changes in the manuscripts.

In figure 3 and its legend, some “gamma-H2AX” is still spelled wrong as y-H2AX.

Reviewer 2 Report

-